# Learning to Predict Task Transferability via Soft Prompt

**Lingyun Feng**
China Mobile Information Technology Center
`fenglingyunit@chinamobile.com`

## Abstract

Fine-tuning pretrained language models on helpful intermediate tasks often greatly improves the performance of target tasks. However, how to efficiently find the source tasks that can successfully transfer still remains under-explored. In this work, we propose to learn an affinity scoring function to predict transferability between tasks. Specifically, we conduct prompt tuning and regard soft prompts as task embeddings that summarize task-specific information. Then we randomly sample task pairs to train an affinity scoring function. The goal is to predict the transfer gain (i.e., affinity) between a task pair, by conditioning on their task embeddings. Once the scoring function is trained, given a novel target task, we use it to predict the most transferable source tasks, without a brute-force search for all possible source-target pairs. Experimental results across 50 tasks show that our method efficiently identifies beneficial tasks for transfer learning.

## 1 Introduction

Fine-tuning pretrained language models (PLMs), such as BERT (Devlin et al., 2018) and RoBERTa (Liu et al., 2019b), achieves remarkable performance on various natural language processing tasks. In addition, fine-tuning on intermediate source tasks yields further gains (Phang et al., 2018; Vu et al., 2020; Pruksachatkun et al., 2020).

While incorporating intermediate stages of knowledge transfer has shown compelling benefits on various tasks, choosing inappropriate source tasks results in negative transfer performance on the target tasks (Pruksachatkun et al., 2020; Bingel and Søgaard, 2017; Chang and Lu, 2021). Previous work use data size, task complexity, or cosine similarity between tasks as well as domains (Vu et al., 2020; Poth et al., 2021; Vu et al., 2021) to model the relationship between datasets. However, the conditions for successful transfer remain unclear (Phang et al., 2018). Given a target task, which source task is the most helpful is not well investigated. How to effectively predict the transferability between tasks is still challenging.

To shed light on the task relationship, in this paper, we conduct a comprehensive study of the transferability between 50 tasks, covering a wide range of text genres and degrees of difficulty. As fine-tuning and deploying a separate instance of the entire large model for each downstream task is prohibitively expensive, we conduct prompt tuning (Liu et al., 2021b,a; Lester et al., 2021) on a diverse set of source tasks. Prompt tuning is a parameter-efficient approach which freezes all parameters of PLMs and merely tunes soft prompts (Lester et al., 2021). As prompts effectively simulate the knowledge of a large language model (Su et al., 2022), we interpret soft prompts as task embeddings following Vu et al. (2021) and store task prompts in a prompt pool. Then we randomly sample task prompt pairs from the pool, along with their transfer gain (i.e., the performance difference between "with transfer" and "without transfer") to learn an affinity scoring function. Once the scoring function is trained, given a novel target task, we use it to predict the most beneficial source task in the prompt pool. Using the affinity scoring function to predict the most helpful source task costs substantially less than using a brute-force search over all possible source-target pairs. Experimental results show that the proposed affinity scoring function efficiently identifies beneficial tasks for transfer learning and obtains larger transfer gain on the target task than baseline methods. Moreover, our method effectively predicts task transferability with much fewer supervision demands.

Our contributions are as follows:

1) We propose an affinity scoring function to learn to predict task transferability and demonstrate that the affinity scoring function can measure task relationship efficiently and ef-

fectively.

2) We conduct an extensive study of the transferability between 50 NLP tasks. We store small learned task-specific prompts in a prompt pool while enabling the reuse of a single frozen pretrained model for all tasks, yielding low computational and storage costs.

3) Experimental results show that the proposed method efficiently identifies beneficial source tasks and predicts task transferability better than existing methods.

## 2 Background: Prompt Tuning

Soft prompts are trainable continuous embeddings prepended to the input. During training, we keep the large pretrained language model frozen and only update the parameters of the prompt. We use soft prompts in our work since discrete prompts need carefully handcrafted design (Gao et al., 2020; Sanh et al., 2021; Bach et al., 2022). Even if significant effort is invested, discrete prompts are likely to be suboptimal (Zhao et al., 2021). It is difficult and time-consuming to finding proper discrete prompts for each task (Liu et al., 2021a; Vu et al., 2021).

Given a series of $K$ input tokens, $x = \{w_1, ..., w_K\}$, where $w_i$ denotes $i$-th token and $K$ denotes the input length. We first utilize PLM to embed the tokens, forming a matrix $X \in \mathbb{R}^{K \times H}$, where $H$ is the dimension of the embedding space. Following reparametrization methods (Li and Liang, 2021; Liu et al., 2021b) which can lead to more stable optimization (Li and Liang, 2021), we use a two-layer feed-forward neural network with parameter $\theta$ as our prompt encoder. Then we use the prompt encoder to embed the randomly initialized prompt tokens, forming a matrix $P \in \mathbb{R}^{L \times H}$, where $L$ denotes the length of the prompt. We concatenate it to the embedded input as $[P; X] \in \mathbb{R}^{(L+K) \times H}$ which then passes through the downstream model as normal while only parameters of the prompt encoder, i.e., $\theta$, are updated. Given the input embedding $X$ and its label $y$, the model is updated by maximizing the likelihood:

$$\max_{\theta} p(y|[P; X]) \qquad (1)$$

Notice that we do not use verbalizers and prompt engineering in our experiments.

## 3 Methodology

As shown in Figure 1, we first construct a prompt pool which consists of a wide range of source task prompts. Then, we randomly sample task pairs from the pool to train a scoring function that learns affinity scores between tasks. Once the affinity scoring function is learned, given a novel target task, we use it to effectively predict the most transferable source task from the pool. Then we initialize the target prompt encoder with the best source prompt and conduct prompt tuning on the target task.

### 3.1 Prompt Pool Construction

Given $n$ source tasks, denoted by $\mathcal{S} = \{\mathcal{S}_1, \mathcal{S}_2, ..., \mathcal{S}_n\}$, we first train a task-specific model for each task in $\mathcal{S}$. Since updating and storing the entire model for each specific task is extremely expensive and the knowledge in a large pretrained language model can be elicited with prompts (Liu et al., 2021a; Li and Liang, 2021; Lester et al., 2021; Ding et al., 2021), we conduct prompt tuning, a lightweight alternative to fine-tuning, on each task and store each task prompt in a prompt pool. Specifically, we first randomly initialize the prompt encoder and learn a task-specific model for each task in $\mathcal{S}$ using Equation (1). We choose the best prompts (according to the validation results) and store them in the prompt pool. Since the task prompts encode task-specific knowledge which is used to reason about the nature of those tasks and their relations (Vu et al., 2021), we interpret the output of the prompt encoder as the task embeddings to construct a semantic space of tasks.

In contrast to full fine-tuning, which updates all transformer parameters and thus requires storing the entire model for each task, we only optimize the prompt parameters and store the prompt encoder (with $\sim 0.1\%$ parameters) for each task in the prompt pool, significantly reducing computing and storage costs.

### 3.2 Task Affinity Scoring Function

We use an affinity scoring function, denoted by $s$, to estimate the transfer gains between tasks, so that we can choose the most helpful source prompt from the constructed pool.

Let $P_1, P_2, ..., P_n$ represent $n$ task embeddings in the prompt pool. We define the transfer gain $a_{ij}$ (from task $i$ to task $j$) as follows: First, we conduct prompt tuning with a randomly initialized

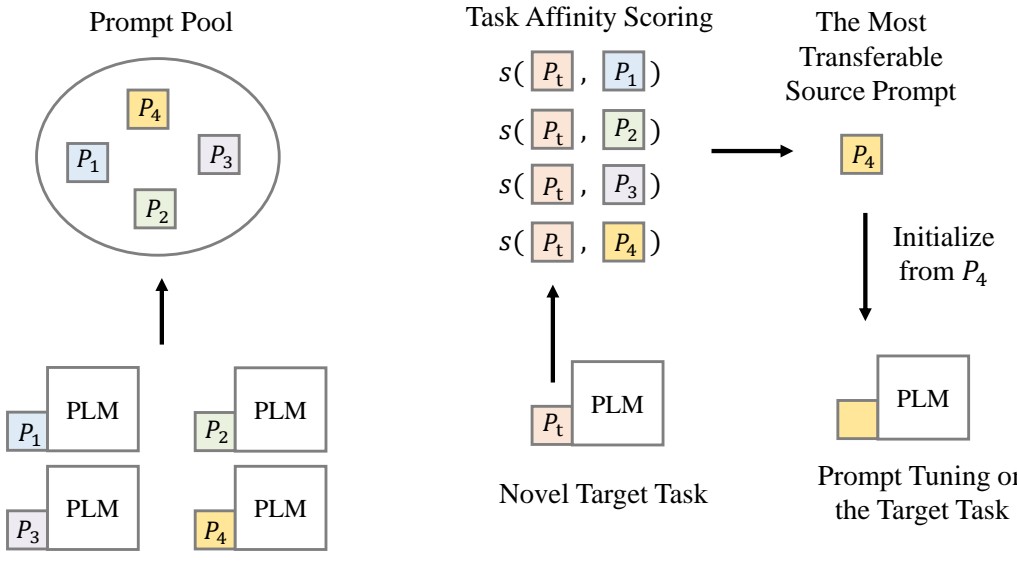

(a) Prompt Pool Construction

(b) Evaluate on Novel Tasks

Figure 1: Overview of the proposed method. We first conduct prompt tuning on a diverse set of source tasks and store each task prompt in the prompt pool. Then, we randomly sample task pairs from the pool to train an affinity scoring function $s$ to estimate the transfer gains between tasks. When the affinity scoring function is well learned, given a novel target task, we first compute its task embedding, then use the affinity scoring function $s$ to select the best source prompt from the prompt pool, and use it to initialize the prompt encoder for the target prompt tuning.

soft prompt to obtain the performance $r_1$ on the task $j$. Second, we employ the task embedding $P_i$ to initialize the prompt for the task $j$, denoting the performance as $r_2$. The transfer gain $a_{ij} = r_2 - r_1$, i.e., the performance difference on the task $j$ between "with transfer" and "without transfer".

We randomly sample $M$ task pairs $< i, j >$ ($\{i, j\} \in [1, n], i \neq j$) from the source pool to train the scoring function $s$. Labels are their transfer gains $a_{ij}$. The backbone network of the affinity scoring function $s$ is initialized from the pretrained model, as language models have demonstrated great power in learning semantic relationship of input representations (Qiu et al., 2020; Zhang et al., 2019). For tasks $i, j$, we concatenate their task embeddings as $[P_j; P_i]$ as the input of the scoring function $s$ and feed it into the scoring function by $s([P_j; P_i])$ in order to predict the transfer gain $a_{ij}$. The training objective is:

$$\min \sum_{i,j} f(a_{ij}, s([P_j; P_i]))$$
$$f(u, v) = \begin{cases} \frac{1}{2}(u-v)^2/\beta & |u-v| \leq \beta \\ |u-v| - \frac{1}{2}\beta & \text{otherwise} \end{cases} \quad (2)$$

where $\beta$ specifies the threshold at which to switch between L1 and L2 loss. We choose smooth L1

loss Baevski et al. (2022) to make training less sensitive to outliers. Once the affinity scoring function $s$ is learned, we use it to effectively predict the transfer gain between tasks.

### 3.3 Evaluation on Novel Target Tasks

Given a novel target task, to find the best source task from the pool to transfer, it is infeasible to conduct an exhaustive search over all possible <source, target> task pairs. In contrast, we use the aforementioned scoring function $s$ to give each task in the prompt pool a task affinity score and take the task with the highest score as the best source task to transfer.

Specifically, given a novel target task $t \in \mathcal{T}$, we first learn target task embeddings $P_t$ using Equation (1). Then we concatenate $P_t$ with each task embedding in the prompt pool $P_i$ ($i \in [1, n]$) as $[P_t; P_i]$. We use the affinity scoring function to give each source task $i$ a task affinity score $s([P_t; P_i])$ ($i \in [1, n]$). We choose the source task that has the highest prediction score, denoted by $P^*$, as the best source task to transfer. The process is formulated as follows:

$$P^* = P_k$$
$$\text{where } k = \arg\max_i (s([P_t; P_i])), i \in [1, n] \quad (3)$$

Then we conduct prompt tuning on the target task $t$ with the prompt encoder initialized by $P^*$. Formally, given the label $y$ and input embedding $X$ of the target task $t$, the model is updated by maximizing the likelihood of the ground-truth $y$:

$$\max_{\theta} p(y|[P^*; X]) \qquad (4)$$

where only the parameters of the prompt encoder, i.e., $\theta$, are updated.

# 4 Experiments

## 4.1 Source and Target Tasks

We experiment on a diverse set of 50 tasks, covering a wide range of text genres and degrees of difficulty. We include natural language inference (NLI) tasks (Williams et al., 2017; Dagan et al., 2005; Haim et al., 2006; Giampiccolo et al., 2007; Bentivogli et al., 2009), paraphrase identification (PI) tasks (Dolan and Brockett, 2005), semantic similarity (Cer et al., 2017), linguistic acceptability (Warstadt et al., 2019), text classification tasks (Socher et al., 2013; Clark et al., 2019) in GLUE (Wang et al., 2018) and SuperGLUE (Wang et al., 2019) benchmarks and other classification tasks (McAuley and Leskovec, 2013; Zhang et al., 2015; De Gibert et al., 2018; Saravia et al., 2018). We also include question answering (QA) tasks (Rajpurkar et al., 2016, 2018; Saha et al., 2018), commonsense reasoning (Bhagavatula et al., 2019), sequence labeling tasks (Pradhan et al., 2012) such as entity recognition (Sang and De Meulder, 2003; Carreras and Màrquez) and chunking (Sang and Buchholz, 2000). We divide them into 44 source tasks and 6 target tasks. Detailed statistics about the tasks are presented in Appendix A.

## 4.2 Implementation Details

We experiment with RoBERTa-large (Liu et al., 2019b) as the pretrained model and set prompt length $L$ to be 50. The prompt encoder is a two-layer feed-forward network and the hidden size is 200. The batch size is set to be 16 and the learning rate is tuned in {1e-4,2e-4}. We use Adam optimizer (Kingma and Ba, 2014), with a linear warmup for the first 6% of steps. For the affinity scoring function $s$, we also use RoBERTa-large model and the learning rate is tuned in {1e-5,2e-5,3e-5}. The task pool has 44 source tasks in total and the number of task pairs $M$ for affinity scoring function learning is 400. The hyperparameter $\beta$

in Equation 2 is set to be 1. Due to the test sets for tasks in GLUE (Wang et al., 2018) and SuperGLUE (Wang et al., 2019) is not publicly available, we evaluate our models on target validation sets. For source and target tasks, the maximum number of samples is set to 50k for the training set and 10k for dev set in consideration of training efficiency.

## 4.3 Baselines

We compare with the following baselines:

- **Brute Force**: We use a brute-force search to identify the best source prompt from the prompt pool to initialize the prompt encoder for the target task.

- **Random Selection**: We randomly choose source task prompts from the prompt pool to initialize the prompt encoder for target tasks.

We also compare with different similarity measurements for task embeddings to estimate the relationship between tasks. Let $P^s, P^t$ denote the source and target task embeddings respectively, $p_i^s, p_j^t$ denote the respective prompt tokens, $L$ denotes the length of the prompt, we measure task similarity as follows:

- **$D_{mah}$**: Manhattan distance of the average representations between source and target task embeddings, i.e., $D_{mah}(P^s, P^t) = \frac{1}{1+|\frac{1}{L}\sum_j p_j^t - \frac{1}{L}\sum_j p_j^t|}$.

- **$\hat{D}_{mah}$**: Per-token average Manhattan distance between the source and target task embeddings, i.e., $\hat{D}_{mah}(P^s, P^t) = \frac{1}{1+\frac{1}{L^2}\sum_i \sum_j |p_i^s - p_j^t|}$.

- **$D_{euc}$** (Su et al., 2021): Euclidean distance of the average representations between source and target task embeddings, i.e., $D_{euc}(P^s, P^t) = \frac{1}{1+||\frac{1}{L}\sum_j p_j^t - \frac{1}{L}\sum_j p_j^t||}$.

- **$\hat{D}_{euc}$** (Su et al., 2021): Per-token average Euclidean distance between the source and target task embeddings, i.e., $\hat{D}_{euc}(P^s, P^t) = \frac{1}{1+\frac{1}{L^2}\sum_i \sum_j ||p_i^s - p_j^t||}$.

- **$D_{cos}$** (Vu et al., 2021): Cosine similarity of the average representations between source and target task embeddings, i.e., $D_{cos}(P^s, P^t) = \frac{\frac{1}{L}\sum_i p_i^s \frac{1}{L}\sum_j p_j^t}{||\frac{1}{L}\sum_i p_i^s||||\frac{1}{L}\sum_j p_j^t)||}$.

- **$\hat{D}_{cos}$** (Vu et al., 2021): Per-token average cosine similarity between the source and target task embeddings, i.e., $\hat{D}_{cos}(P^s, P^t) = \frac{1}{L^2} \sum_i \sum_j \frac{p_i^s p_j^t}{||p_i^s|| ||p_j^t||}$.

We use the aforementioned different methods to measure the similarity between task embeddings. The source prompt whose associated task embedding has the highest similarity to the target embedding is chosen as the best source prompt. Then we use the best source prompt to initialize the target prompt encoder and conduct prompt tuning on the target task. For fair comparisons, experiment settings except the source prompt choice are set to be the same.

### 4.4 Results

We compare different methods of predicting the most beneficial source task given a novel target task and report their performance on each target task in Table 1. We first conduct a brute force search with all source and target task combinations. Experimental results demonstrate that the choice of intermediate source tasks can heavily affect target task performance[1], e.g., improve the performance of MRPC target task by 8.83% and degrade the performance on the SST-2 target task by almost 10%. On average, a brute-force search for the best source task has 4.29% accuracy improvements over no transfer baseline.

In general, tasks with fewer training samples benefit the most from transfer learning, e.g., 8.28% accuracy improvements on CoNLL2004 and 8.83% accuracy improvements on MRPC task. It shows the importance and necessity of task transferability prediction. Transferring from useful intermediate source tasks can provide significant gain on the target task.

We also observe that positive transfer can occur when the source task has a small size or a different task type, e.g., question answering tasks such as SQuAD produce high transfer gain on semantic similarity task QQP, CoNLL2000 task which has a small amount of data improves the performance on MRPC task by a large margin. It indicates that traditional methods such as using data size or domain similarity fail to estimate task affinity precisely.

When adopting different transferability metrics to choose source tasks, experimental results show

---

[1]The detailed results of all source and target combinations and transfer gains on each target task with different source tasks in the pool are presented in Appendix B

that using Manhattan distance and Euclidean distance for task similarity computation always have the same transfer gain on the target task, as these methods tend to predict the same source task to transfer. However, these heuristic methods can not measure the task similarity well, and even degrade the performance on the target tasks (e.g., -2.35% on average for $D_{mah}$). Using cosine similarity to measure task relationship performs better but still achieves sub-optimal performance on the target task.

Compared with other similarity measurements, the affinity scoring function measures the task transferability more effectively and achieves the highest average transfer gains on the target task. It improves the performance on the target task by 3.78% on average, close to the upper bound results with brute-force search which is 4.29%.

Besides, we also evaluate the correlation between task transferability and task similarity measured by the aforementioned methods. Given a target task, we rank all the source tasks in the prompt pool by different similarity measurements. The ranking is evaluated using three metrics: Pearson correlation, Spearman correlation, and Regret@k (Renggli et al., 2020) which measures the relative performance difference between the Top-k selected source tasks and the optimal one following Poth et al. (2021). Results are shown in Table 2, using Manhattan distance and Euclidean distance achieves similar but sub-optimal performance. Using per-token average cosine similarity between the source and target task embeddings performs better than calculating the cosine similarity of the average representations to measure task similarity. However, none of them can effectively learn the task relationship. The proposed affinity scoring function correlates better with task transferability and provides higher transfer gain on the target task than existing methods. It also demonstrates that the soft prompt effectively encodes task-specific characteristics and serves as a good means to explore task transferability.

### 4.5 Ablation Studies

**Effect of $\beta$.** We study the effect of $\beta$ in Equation (2) which controls the transition from a squared loss to an L1 loss. We vary $\beta \in \{0.5, 1, 1.5, 2, 2.5\}$ and report the Spearman correlation between the prediction of the affinity scoring function and the ground-truth transfer gains. As

| | SST-2 | QQP | MRPC | IMDB | SciTail | CoNLL2004 | MEAN($\triangle$) |
|---|---|---|---|---|---|---|---|
| *Upper bound results of task transfer with brute-force search* | | | | | | | |
| Brute Force | 96.22 | 86.56 | 90.69 | 94.78 | 97.09 | 95.31 | 4.29 |
| *Baseline results without transfer or using random source task for transfer* | | | | | | | |
| No Transfer | 95.07 | 83.94 | 81.86 | 93.90 | 93.10 | 87.03 | - |
| Random Selection | 84.29 | 84.79 | 79.90 | 93.68 | 95.78 | 84.78 | -1.95 |
| *Use transferability metrics to choose source tasks* | | | | | | | |
| $D_{\text{man}}$ | 94.15 | 83.35 | 78.19 | 93.76 | 85.97 | 85.37 | -2.35 |
| $D_{\text{euc}}$ (Su et al., 2021) | 94.15 | 83.35 | 78.19 | 93.76 | 89.80 | 85.37 | -1.71 |
| $D_{\text{cos}}$ (Vu et al., 2021) | **96.22** | 85.83 | 78.19 | **94.58** | 94.87 | **95.31** | 1.68 |
| $\hat{D}_{\text{man}}$ | 94.15 | 83.35 | 78.19 | 93.76 | 89.80 | 85.37 | -1.71 |
| $\hat{D}_{\text{euc}}$ (Su et al., 2021) | 94.15 | 83.35 | 78.19 | 93.76 | 89.80 | 85.37 | -1.71 |
| $\hat{D}_{\text{cos}}$ (Vu et al., 2021) | 95.41 | 85.58 | 78.19 | **94.58** | 96.70 | **95.31** | 1.81 |
| Affinity Scoring Function | 95.53 | **86.25** | **89.46** | 94.48 | **97.09** | 94.75 | **3.78** |

Table 1: Accuracy on the target tasks. All values are scaled by 100. "No Transfer" shows the baseline performance when training on the target task with randomly initialized prompt without transfer. We also report the average absolute transfer gain in the rightmost column of the table. A positive delta indicates successful transfer.

| | S. (↑) | P. (↑) | R@1 (↓) | R@3 (↓) |
|---|---|---|---|---|
| $D_{\text{man}}$ | 0.45 | 0.39 | 1.55 | 1.17 |
| $D_{\text{euc}}$ | 0.45 | 0.40 | 1.40 | 1.17 |
| $D_{\text{cos}}$ | 0.26 | 0.22 | 0.61 | 0.61 |
| $\hat{D}_{\text{man}}$ | 0.33 | 0.29 | 1.40 | 1.17 |
| $\hat{D}_{\text{euc}}$ | 0.33 | 0.30 | 1.40 | 1.17 |
| $\hat{D}_{\text{cos}}$ | 0.39 | 0.41 | 0.58 | 0.51 |
| Affinity Scoring Function | **0.55** | **0.45** | **0.12** | **0.06** |

Table 2: Evaluation of intermediate source task rankings produced by different methods. S. and P. are short for Spearman correlation and Pearson correlation, respectively. R@k is short for Regret@k. For Spearman and Pearson correlation, higher is better; for Regret@k, lower is better.

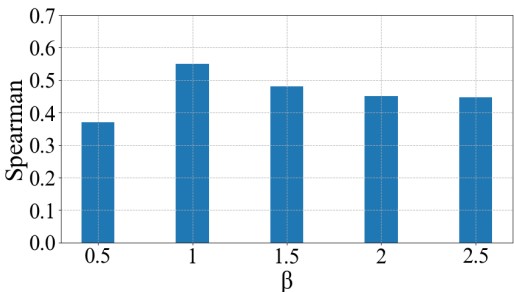

Figure 2: Effect of $\beta$ on affinity scoring function learning. The y-coordinate denotes the Spearman correlation between the prediction of the affinity scoring function and the ground-truth transfer gains.

shown in Fig 2, a larger $\beta$ improves the affinity scoring function, while too large values tend to harm results. The best performance is achieved when $\beta = 1$.

### 4.6 Analysis

#### 4.6.1 Efficiency

To demonstrate the efficiency of the proposed affinity scoring function, we vary the number of selected task pairs $M \in \{100, 200, 300, 400\}$ and use them to train the affinity scoring function respectively. Then we use the affinity scoring function trained with the respective amount of supervision signal to access tasks in the prompt pool and select best of Top-$k$ ($k \in \{1, 3\}$) source prompt to initialize the target prompt encoder. We compare the proposed affinity scoring function with the most competitive

baselines: $\mathbf{D_{cos}}$ and $\hat{\mathbf{D}}_{\mathbf{cos}}$ in Vu et al. (2021).

Results are shown in Figure 3. Though using cosine similarity as task similarity measurement improves the performance of the model on the target task, the affinity scoring function consistently performs better than them by a large margin. Even when the number of task pairs to train the affinity scoring function is 100, which is much fewer than the total task pairs ($44 \times 44 = 1936$ task pair combinations in total), the affinity scoring function still outperforms the baseline methods by 1.54% on average for best of Top-1 and 1.75% for best of Top-3 setting. As the number of the training task pairs increases, the affinity scoring function performs better. When there are only 400 training task pairs, the affinity scoring function achieves 4.01% gain in accuracy with the best of Top-3 source prompt while the oracle selection of source

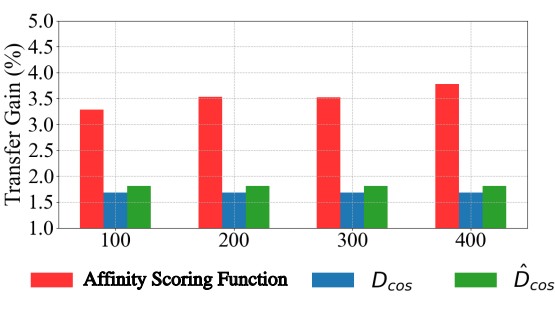

(a) Best of Top-1

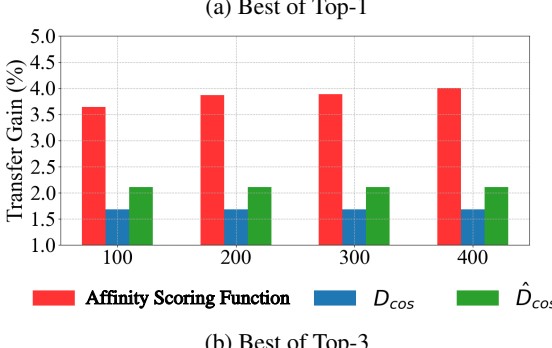

(b) Best of Top-3

Figure 3: Average absolute transfer gain on the target tasks with the Best of Top-1 and Top-3 source prompt chosen by different methods.

prompts (exhaustively brute force fine-tuning on all possible source-target pairs) is 4.29%. It indicates that with randomly selected few task pairs from the prompt pool, the proposed affinity scoring function is able to effectively predict the task transferability, rather than enumerate every possible task combination pair. Given a novel target task, the affinity scoring function efficiently predicts the most beneficial source tasks with much fewer supervision demands.

### 4.6.2 Generalization

To investigate the generalization of the affinity scoring function on new task types [2], we remove the sequence labeling tasks (including named entity recognition, word segmentation, chunking, and part of speech) from the pool, using the remaining tasks to train the affinity scoring function and test its prediction performance on the new task CoNLL2004 (named entity recognition task). Experimental results show that the affinity scoring function is still able to select beneficial source tasks from the pool and achieve positive transfer gain on the target task: the largest transfer gain on the CoNLL2004 task is 5.71% in accuracy. It suggests that the affinity scoring function is generalizable to different task types.

---

[2]Detailed task division by type is presented in Appendix A

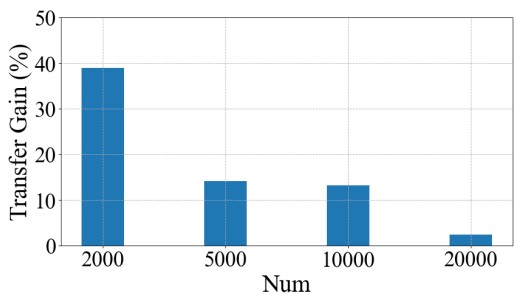

Figure 4: Absolute transfer gain on the target task with different target datasize.

### 4.6.3 Low-Resource Setting

To investigate how the affinity scoring function performs when the target task is in the low-resource scenario, we take SST-2 task as the target task and subsample the full datasets to obtain small sets of size $\{2000, 5000, 10000, 20000\}$. We use the proposed affinity scoring function to select the most beneficial source task, i.e., MNLI task, to initialize the target prompt encoder and conduct prompt tuning on SST-2 in the settings of different data sizes. We report the absolute transfer gain on SST-2 target task for each data size in Figure 4. We observe that the proposed affinity scoring function yields larger transfer learning improvement when the target data is in a low-data regime. Though the model achieves low accuracy for data-constrained target tasks, the performance is significantly improved with the help of transfer learning (38.99% with 2000 training samples). It also echoes that effective task transferability prediction greatly reduces the supervision cost on the target task, which can be particularly valuable when there is insufficient data to train or finetune the model.

### 4.6.4 Intermediate-Task Full Fine-Tuning Setting

The task relationship predicted by the affinity scoring function is helpful not only for prompt transfer learning but also for intermediate-task full fine-tuning setting. To demonstrate this, we first randomly select 20 source tasks from the pool and test their transfer performance on the target in-pool task (e.g., RTE task) and out-of-pool task (e.g., MRPC task) respectively with different fine-tuning methods. We find that the transfer gain obtained by prompt tuning is positively correlated with the transfer gains achieved by full fine-tuning, as the Spearman correlation is 0.13 for MRPC task and 0.54 for RTE task. Furthermore, we choose the most transferable source task predicted by the affin-

| Method | SST-2 | MRPC | QQP | SciTail | IMDB | CoNLL2004 |
|---|---|---|---|---|---|---|
| No Tranfer | 96.10 | 88.97 | 88.43 | 97.32 | 94.96 | 95.57 |
| Intermediate Fine-Tuning | 96.22 | 91.91 | 88.76 | 97.39 | 95.06 | 96.20 |
| Δ | +0.12 | +2.94 | +0.33 | +0.07 | +0.10 | +0.63 |

Table 3: Accuracy on the target tasks with vanilla fine-tuning and intermediate fine-tuning. All values are scaled by 100. We also calculate the absolute transfer gain obtained by transfer learning on these target tasks.

ity scoring function and conduct full fine-tuning on the six target tasks respectively, following sequential fine-tuning setup in Phang et al. (2018), i.e., first fine-tune on the chosen intermediate source task, then fine-tune the model on the specific target task. Table 3 shows the absolute transfer gain on the target tasks with intermediate source task fine-tuning. We find that the source tasks selected by the affinity scoring function achieve consistent gains on the target tasks in full fine-tuning setting. The transfer gain is more obvious for low-resource target tasks, e.g., it achieves 2.94% gain in accuracy on the MRPC task. It indicates that the proposed affinity scoring function is capable of choosing beneficial source tasks for the given target task not only in the parameter-efficient setting but also in full model fine-tuning.

## 5   Related work

**Transfer between tasks** has been investigated in many research areas, such as NLP (Poth et al., 2021; Bingel and Søgaard, 2017; Pruksachatkun et al., 2020; Vu et al., 2020; Liu et al., 2019a; Li et al., 2022) and computer vision (Zamir et al., 2018; Achille et al., 2019; Wang et al., 2022). Li et al. (2022) propose a prompt-based method in a transferable setting for text generation tasks while we focus on natural language understanding (NLU) tasks. Phang et al. (2018) explore whether fine-tuning on intermediate source tasks can further improve the performance on the target tasks for text classification. Talmor and Berant (2019) conduct an empirical investigation of generalization and transfer in reading comprehension. Vu et al. (2020) compute a task embedding based on the model's gradients with respect to the task-specific loss while Poth et al. (2021) focus on adapter-based intermediate fine-tuning. To investigate task transferability in a more parameter-efficient way, Vu et al. (2021) propose a prompt-based transfer learning approach and cast task prompts as task embeddings which further reduce the computational costs. They use cosine similarity to select the best

source tasks given the target task. Compared with them, we adopt an affinity scoring function which is more effective and efficient to predict the task transfer gain. Besides, the proposed method is not only helpful for prompt transfer learning, but also can be generalized to intermediate-task transfer as well.

**Prompt tuning** is a parameter-efficient method that only tunes "soft prompts" with frozen language models. In contrast to discrete prompts with carefully handcrafted design (Brown et al., 2020; Schick and Schütze, 2020; Jiang et al., 2020; Sanh et al., 2021; Bach et al., 2022), soft prompts are trainable continuous embeddings to the original sequence of input word embeddings which can be learned through back-propagation (Liu et al., 2021b; Wang et al., 2021; Lester et al., 2021; Schick and Schütze, 2020; Wu and Shi, 2022; Asai et al., 2022a). Lester et al. (2021) learns task-specific soft prompts and shows that prompt tuning can be comparable to fine-tuning when the model exceeds billions of parameters. Gu et al. (2021) explore the effectiveness of prompt pre-training. Vu et al. (2021) cast each task into a unified text-to-text format and propose a prompt-based transfer learning approach with T5 model. Wang et al. (2022) learns to dynamically prompt a pre-trained model to learn tasks sequentially for computer vision tasks. Liu et al. (2021a) adapt prefix-tuning (Li and Liang, 2021) for natural language understanding. They apply continuous prompts for every layer of pretrained models, showing that prompt tuning can be comparable to fine-tuning universally across scales and tasks. Asai et al. (2022b) proposes parameter-efficient multi-task tuning via attentional mixtures of soft prompts, which requires high-resource source datasets and multiple target tasks. By contrast, in our work, the target can be from one or more tasks and the source data can be less sufficient. Besides, the proposed method is helpful not only for prompt transfer learning, but also for intermediate-task transfer.

# 6 Conclusions

In this paper, we conduct an extensive study of the transferability between 50 tasks and propose an affinity scoring function to predict transferability between tasks. We interpret soft prompts as task embeddings, and store task prompts in a prompt pool. The affinity scoring function is trained with task pairs sampled from the pool to learn to estimate task affinity. Then given a novel target task, we use the affinity scoring function to predict helpful source tasks from the pool. Experimental results show that the proposed method outperforms the baseline methods on task affinity prediction. Moreover, our method effectively predicts task transferability with much fewer supervision demands.

## Limitations

Given a novel target task, we predict the most beneficial source task and show that fine-tuning on a helpful intermediate source task can greatly improve the performance on the target task. In this paper, we use one source task prompt to initialize the prompt encoder for the target task. We conduct single intermediate-task training to study the affinity between intermediate source tasks and target tasks. It is possible that the performance on the target task can be further improved by properly combining multiple source tasks. We will explore mixing up prompts from multiple source tasks for intermediate-task transfer learning for future work.

## Acknowledgements

This work is supported by Big Data Key Software System Research and Development Project (R23100LX).

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

# Appendix

## A  Datasets

Detailed statistics of source and target tasks are presented in Table 4.

## B  Transfer Results

Detailed transfer results on target tasks with different source tasks are presented in Table 5. We further group source tasks by different task types and show the transfer gains on each target task in Fig. 5.

| Task | \|Train\| | \|Dev\| | Task Type | Metrics |
|------|--------|------|-----------|---------|
| Cosmos QA (Huang et al., 2019) | 25.3k | 3k | Commonsense Reasoning | ACC |
| SWAG (Zellers et al., 2018) | 50k | 10k | Commonsense Reasoning | ACC |
| DuoRC-s (Saha et al., 2018) | 50k | 10k | QA | F1 |
| DuoRC-p (Saha et al., 2018) | 50k | 10k | QA | F1 |
| SICK (?) | 4.4k | 0.5k | NLI | ACC |
| TREC (?) | 4.4k | 1k | Question Classification | ACC |
| SciCite (Cohan et al., 2019) | 8.2k | 1k | Citation Intent Classification | ACC |
| CoLA (Warstadt et al., 2019) | 8.6k | 1k | Linguistic Acceptability | MCC |
| Emotion (Saravia et al., 2018) | 16k | 2k | Emotion Classification | ACC |
| IMDB (Maas et al., 2011) | 20k | 5k | Sentiment Classification | ACC |
| Rotten Tomatoes (Pang and Lee, 2005) | 8.5k | 1.1k | Sentiment Classification | ACC |
| STS-B (Cer et al., 2017) | 5.7k | 1.5k | Semantic Textual Similarity | Pearson |
| Yelp Polarity (Zhang et al., 2015) | 50k | 10k | Sentiment Classification | ACC |
| SNLI (Bowman et al., 2015) | 50k | 9.8k | NLI | ACC |
| MNLI (Williams et al., 2017) | 50k | 9.8k | NLI | ACC |
| QQP [3] | 50k | 10k | Semantic Textual Similarity | ACC |
| QNLI (Wang et al., 2018) | 50k | 5.5k | NLI | ACC |
| SST-2 (Socher et al., 2013) | 50k | 0.9k | Sentiment Classification | ACC |
| SciTail (Khot et al., 2018) | 27k | 1.3k | NLI | ACC |
| SQuAD 1.0 (Rajpurkar et al., 2016) | 50k | 10k | QA | F1 |
| SQuAD 2.0 (Rajpurkar et al., 2018) | 52.2k | 10.6k | QA | F1 |
| NER-WNUT17 (Derczynski et al., 2017) | 3.4k | 1k | NER | ACC |
| Chunk-CoNLL2000 (Sang and Buchholz, 2000) | 7.1k | 1.8k | Chunking | ACC |
| POS-CoNLL2003 (Sang and De Meulder, 2003) | 14k | 3.3k | POS | ACC |
| NER-CoNLL2003 (Sang and De Meulder, 2003) | 14k | 3.3k | NER | ACC |
| ST-PMB (Abzianidze and Bos, 2017) | 50k | 10k | Semantic Tagging | ACC |
| NER-MIT Movie | 6.3k | 1.6k | NER | ACC |
| FCE-error-detection (Rei and Yannakoudakis, 2016) | 2.9k | 2.2k | Error Detection | ACC |
| BoolQ (Clark et al., 2019) | 9.4k | 3.3k | QA | ACC |
| CB (De Marneffe et al., 2019) | 250 | 57 | NLI | ACC |
| COPA (Gordon et al., 2012) | 400 | 100 | Commonsense Reasoning | ACC |
| MultiRC (Khashabi et al., 2018) | 27k | 4.8k | QA | F1_a |
| RTE (Dagan et al., 2005) | 2.5k | 277 | NLI | ACC |
| WiC (Pilehvar and Camacho-Collados, 2018) | 5.4k | 638 | WSD | ACC |
| MRPC (Dolan and Brockett, 2005) | 3.7k | 408 | Semantic Textual Similarity | ACC |
| CoNLL2004 (Carreras and Màrquez) | 922 | 231 | NER | ACC |
| AG news (Zhang et al., 2015) | 50k | 10k | Topic Classification | ACC |
| Amazon Polarity (McAuley and Leskovec, 2013) | 50k | 10k | Sentiment Classification | ACC |
| Hate-speech18 (De Gibert et al., 2018) | 8.8k | 2189 | hatespeech identification | ACC |
| Hate-speech-offensive (Davidson et al., 2017) | 19.8k | 5k | Hatespeech Identification | ACC |
| CoNLL2012 (Pradhan et al., 2012) | 50k | 10k | POS | ACC |
| ATIS (Hemphill et al., 1990) | 3982 | 996 | Intent Classification | ACC |
| CR (Ding et al., 2008) | 2.9k | 719 | Sentiment Classification | ACC |
| ontonotes (Weischedel et al., 2013) | 50k | 8.5k | NER | ACC |
| DBpedia14 (Lehmann et al., 2015) | 50k | 10k | Ontology Classification | ACC |
| Laptop14 (Pontiki et al., 2016) | 1.7k | 432 | Sentiment Classification | ACC |
| Emo (Chatterjee et al., 2019) | 24k | 6k | Emotion Classification | ACC |
| ART (Bhagavatula et al., 2019) | 50k | 1.5k | Commonsense Reasoning | ACC |

Table 4: Detailed statistics about the source and target tasks. WSD stands for Word Sense Disambiguation task. QA stands for question answering task. NLI stands for natural language inference task. NER stands for Named Entity Recognition task. POS stands for part-of-speech task. MCC is short for Matthews Correlation Coefficient and ACC stands for accuracy.

|  | SST-2 | MRPC | QQP | SciTail | IMDB | CoNLL2004 |
|---|---|---|---|---|---|---|
| No transfer | 95.07 | 81.86 | 83.94 | 93.1 | 93.9 | 87.03 |
| AG News | 95.18 | 81.86 | 83.80 | 93.25 | 93.96 | 92.06 |
| Amazon Polarity | 95.41 | 86.76 | 84.79 | 94.63 | 94.30 | 90.73 |
| APP Reviews | 84.29 | 79.17 | 77.43 | 93.48 | 86.70 | 84.01 |
| ART | 95.18 | 87.99 | 84.19 | 95.78 | 93.08 | 84.36 |
| ATIS | 92.43 | 82.11 | 84.07 | 92.79 | 92.90 | 89.09 |
| BoolQ | 94.84 | 71.81 | 78.17 | 71.55 | 89.38 | 87.26 |
| CB | 86.47 | 79.41 | 82.17 | 87.27 | 87.88 | 89.02 |
| COLA | 94.72 | 82.84 | 83.55 | 93.87 | 93.50 | 84.27 |
| CoNLL2000 | 95.30 | 90.69 | 85.63 | 96.63 | 94.56 | 94.75 |
| NER-CoNLL2003 | 95.30 | 87.25 | 85.67 | 96.17 | 94.40 | 95.31 |
| POS-ConNLL2003 | 95.07 | 83.58 | 85.33 | 95.78 | 94.56 | 93.09 |
| CoNLL2012 | 94.50 | 88.73 | 85.70 | 96.09 | 94.06 | 94.22 |
| COPA | 94.15 | 78.19 | 83.35 | 89.80 | 93.76 | 85.37 |
| Cosmos QA | 95.30 | 89.22 | 85.58 | 97.09 | 94.78 | 92.74 |
| CR | 95.07 | 82.84 | 84.36 | 95.55 | 93.68 | 88.30 |
| DBpedia_14 | 94.72 | 86.27 | 85.30 | 95.71 | 94.30 | 91.35 |
| DuoRC_p | 95.99 | 90.20 | 86.10 | 96.17 | 94.48 | 94.29 |
| DuoRC_s | 95.64 | 89.46 | 86.16 | 96.63 | 94.64 | 93.32 |
| Emo | 95.07 | 83.82 | 85.42 | 92.95 | 94.12 | 88.27 |
| Emotion | 93.81 | 78.19 | 84.55 | 93.63 | 87.36 | 88.65 |
| FCE-error-detection | 95.76 | 86.27 | 85.09 | 95.63 | 94.50 | 91.66 |
| Hate_speech18 | 83.03 | 78.43 | 84.20 | 91.79 | 85.54 | 86.23 |
| hate_speech_offensive | 88.76 | 83.82 | 83.36 | 88.11 | 91.22 | 84.78 |
| Laptop14 | 93.58 | 81.37 | 83.15 | 92.48 | 87.30 | 89.80 |
| MIT movie | 96.10 | 88.24 | 85.57 | 96.01 | 93.72 | 94.51 |
| MNLI | 95.53 | 89.22 | 86.32 | 94.87 | 94.24 | 90.26 |
| MultirRC | 94.38 | 72.30 | 82.59 | 85.97 | 93.68 | 84.84 |
| ontonotes | 95.07 | 86.76 | 85.85 | 96.47 | 94.34 | 95.05 |
| ST-PMB | 95.76 | 72.06 | 85.35 | 96.47 | 94.72 | 93.71 |
| QNLI | 95.41 | 83.82 | 85.83 | 93.72 | 93.84 | 88.15 |
| QuAIL | 93.81 | 80.64 | 83.61 | 93.02 | 93.88 | 84.93 |
| Rotten Tomatoes | 96.22 | 83.33 | 85.35 | 96.32 | 94.28 | 88.57 |
| RTE | 90.71 | 79.17 | 83.42 | 90.72 | 85.28 | 81.87 |
| SciCite | 95.18 | 84.56 | 85.22 | 93.49 | 94.12 | 90.30 |
| SICK | 96.10 | 87.25 | 85.52 | 93.72 | 93.98 | 86.41 |
| SNLI | 96.21 | 88.97 | 85.85 | 94.10 | 93.88 | 90.75 |
| SQuAD | 95.99 | 89.71 | 86.25 | 95.94 | 94.58 | 93.55 |
| SQuAD_v2 | 95.64 | 90.44 | 86.56 | 96.70 | 94.54 | 92.56 |
| STS-B | 94.61 | 85.05 | 84.55 | 94.71 | 86.58 | 88.59 |
| SWAG | 93.35 | 73.04 | 84.19 | 93.25 | 93.40 | 89.76 |
| TREC | 95.18 | 82.35 | 85.89 | 93.41 | 94.02 | 89.93 |
| WiC | 95.64 | 79.90 | 84.55 | 94.79 | 93.28 | 88.86 |
| NER-WNUT17 | 95.64 | 77.70 | 85.52 | 94.56 | 94.58 | 94.75 |
| Yelp Polarity | 95.41 | 84.56 | 85.49 | 95.78 | 94.58 | 87.29 |

Table 5: Accuracy on each target task with different source tasks. All values are scaled by 100.

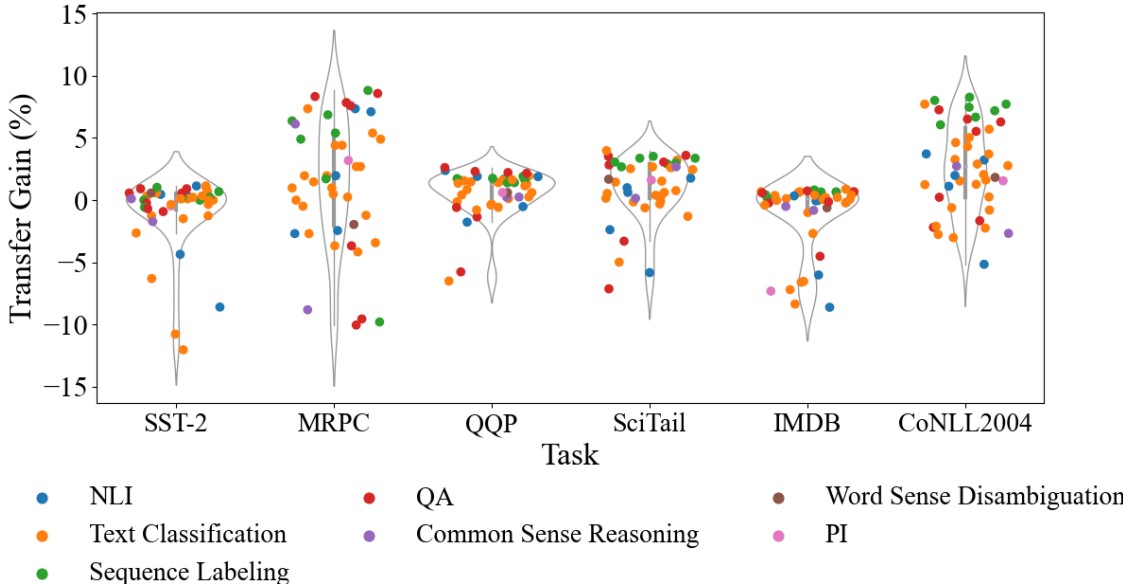

Figure 5: Transfer gains on each target task with different source tasks. Each violin represents one target task. Each point inside a violin represents an individual source task. The point color denotes task type, and the y-coordinate denotes the absolute transfer gains on the specific target task with source tasks. Here we group named entity recognition, word segmentation, chunking, and part of speech into sequence labeling tasks; sentiment classification, emotion classification, hate speech identification, question classification, intent classification, grammatical judgment, and topic classification into text classification tasks.