# OpenReview forum: "Learning to Predict Task Transferability via Soft Prompt"
_EMNLP/2023/Conference — EMNLP 2023 Main_

### Official Review · Reviewer_fsyc · 2023-08-02

**Soundness:** 3

**Excitement:**

3: Ambivalent: It has merits (e.g., it reports state-of-the-art results, the idea is nice), but there are key weaknesses (e.g., it describes incremental work), and it can significantly benefit from another round of revision. However, I won't object to accepting it if my co-reviewers champion it.

**Paper Topic And Main Contributions:**

This paper studies a method to predict the transferability between tasks through learning an affinity scoring function.

The contributions of this paper are twofold. First, this work proposes to learn an affinity scoring function to predict the transferability between 50 tasks. Second, the extensive experiments validate the effectiveness of this method to some extent.

**Questions For The Authors:**

1. In Section 3.2, you claimed that the task embedding P_i is used to initialize the prompt for task j. In this way, P_i will be trained using the training data of task j "again" after learning P_j. Therefore, every calculation of transfer gain involves two times of training, which will lead to high computation costs. Can you provide the total training time in your experiments?
2. You are missing some important baselines including GPT-3 and ChatGPT. Since today's researches are mostly focused on large language models using discrete prompts (or instructions), you should compare with them and discuss your advantages on them.
3. For table 1, you should list or present the performance of Affinity Scoring Function is based on which source task, which will make your analysis in Section 4.4 more clear.
4. I understood that you only use the prompt of the best source task. So in Section 4.6.1, is there any difference for k=1 or 3?
5. This work is very similar to previous work [1,2,3], but you miss explicit and insightful comparison with them.
[1] Learning to transfer prompts for text generation;
[2] Learning to prompt for continual learning;
[3] ATTEMPT: Parameter-Efficient Multi-task Tuning via Attentional Mixtures of Soft Prompts.

**Reasons To Accept:**

1. Novelty in method. This work is the first to leverage an affinity scoring function to measure the task transferability.
2. Extensive experiments. This work conducts many related experiments.
3. The writing is clear to understand this paper.

**Reasons To Reject:**

1. There are some drawbacks of this method such as training complexity.
2. Missing some important baselines and not clear experimental analysis.
3. Missing insightful comparison with missing important references.


**Reproducibility:**

4: Could mostly reproduce the results, but there may be some variation because of sample variance or minor variations in their interpretation of the protocol or method.

**Reviewer Confidence:**

5: Positive that my evaluation is correct. I read the paper very carefully and I am very familiar with related work.

---

> ### Author Rebuttal · Authors · 2023-08-27
>
> Thanks for your valuable suggestions!
>
> 1. Every calculation of transfer gain only cost a few minutes with one GPU(V100).  Rather than enumerating every task combination pair, we randomly selected 400 task pairs from the prompt pool to calculate the transfer gain. Thus, the total training time is around 16 hours with 8 V100 GPUs. We will add the total training time in the revised version.
>
> 2. Discrete prompts need carefully handcrafted design. Even if significant effort is invested, discrete prompts are likely to be suboptimal[4]. It is difficult and time-consuming to finding proper prompts for each task[5][6]. So we use soft prompt in the training process. We will add the comparison of soft prompts and discrete prompts in the revised version.
>
> 3. Thanks for your suggestion! We will list the source task which the performance of Affinity Scoring Function is based on in the revised version.
>
> 4. Given a novel target task, the affinity function gives each task in the prompt pool a task affinity score. For k=1, we take the task with the highest score as the best source task to transfer. For k=3, we select the source tasks with top-3 scores, conduct prompt tuning using these three source tasks respectively and present the best performance on the the target task (i.e., selecting the best of top-k requires prompt tuning k times on the target task). We report the average absolute transfer gain on the target tasks with the Best of Top-1 and Top-3 source prompt in Fig.3.
>
> 5. [2] learns to dynamically prompt a pre-trained model to learn tasks sequentially for computer vision tasks, [1] learns to transfer prompts for text generation tasks, while we focus on natural language understanding (NLU) tasks. [3] proposes parameter-efficient multi-task tuning via attentional mixtures of soft prompts. Our work differs from theirs as follows:
>
>       (1) They conduct multi-task learning over a group of target tasks by sharing the attention module. Different from [3] which requires high-resource source datasets and multiple target tasks, in our work, the target can be from one or more tasks and the source data can be less sufficient.
>
>       (2) They focus on classification and QA tasks while our method can be generalizable to more diverse sets of NLU tasks such as sequence labeling tasks.
>
>       (3) The proposed method is not only helpful for prompt transfer learning, but also can be generalized to intermediate-task transfer as well.
>
> [1] Learning to transfer prompts for text generation. Li et al. 2022.
>
> [2] Learning to prompt for continual learning. Wang et al. 2022
>
> [3] ATTEMPT: Parameter-Efficient Multi-task Tuning via Attentional Mixtures of Soft Prompts. Asai et al.2022.
>
> [4] Calibrate before use: Improving few-shot performance of language models. Zhao et al., 2021.
>
> [5] P-Tuning v2: Prompt Tuning Can Be Comparable to Fine-tuning Universally Across Scales and Tasks. Liu et al.,2021
>
> [6] SPoT: Better Frozen Model Adaptation through Soft Prompt Transfer. Vu et al.,2021.

---

### Official Review · Reviewer_23Bc · 2023-08-02

**Soundness:** 4

**Excitement:**

4: Strong: This paper deepens the understanding of some phenomenon or lowers the barriers to an existing research direction.

**Paper Topic And Main Contributions:**

This paper deals with the topic of choosing the optimal source task when conducting transfer learning to a target task. While it is well known that knowledge transfer benefits pretrained language models, choosing a sub-optimal source task will either not benefit or may even harm performance on the target task. However, methods of finding optimal source tasks remain underexplored, and brute-force search over tasks is inefficient. Thus, it is relevant to investigate efficient methods of finding source tasks that will result in performance gains (i.e. tasks that are transferable).

This paper provides a method of predicting the transferability of tasks by employing an affinity-scoring function based on soft prompts. First, source task prompts are collected into a prompt pool. Next, an affinity scoring function is learned, which represents the transfer gain (gain in performance after transfer) for each source task with the given target task. The source task with the highest affinity score is chosen for transfer. The performance is evaluated against several baseline methods, including brute-force search, random selection, and various similarity measures. It is found that the proposed affinity scoring function method outperforms all baselines in average absolute transfer gain save for brute-force search.

Concretely, the paper makes the following contributions:

1) A new affinity scoring function based on soft prompts that determines which source task is the most beneficial for transfer to a target task.
2) An ablation study and experiments that demonstrate the effectiveness of the proposed scoring function.

**Questions For The Authors:**

A) Line 57: The phrase "simulate the knowledge of a large language model" is unclear. Do you mean that prompts effectively enable LLMs to access their knowledge? (This is what I would expect.)

B) Section 2: The prompt tuning procedure is slightly unclear. What do you mean by "randomly-initialized prompt tokens" (lines 109-110)? Does this mean that the initial prompt for each task consists of random tokens? Why not start with a prompt that would seem good from a human perspective, even though it's not engineered/optimized?

C) Lines 134-135: What kind of architecture is being used as the task-specific model here? Is it an LLM?

D) Lines 343-350: These two sentences seem to contradict each other. The first sentence ("We also observe that positive transfer...") states that positive transfer can occur when the source task is small, which implies that small task size is a potential indicator of good transferability. But the final sentence ("It indicates that...") seems to say that data size can't be used as a predictor of task affinity. Didn't the observations indicate otherwise?

E) Are the scores in Table 1 the average of multiple runs, or only from single runs?

**Reasons To Accept:**

The task of determining the best source task for transfer is important and nontrivial. This paper proposes an effective method for doing so that requires less supervision than previous methods. The paper's theory is adequately explained by equations, and the experiments provide an extensive comparison with 2 baseline selection methods, as well as 6 baseline similarity measures. This provides a good justification of the effectiveness of the scoring function, and in fact the scoring function outperforms all baselines, save for brute force search (which is expensive). Finally, an ablation study is performed to test the optimal parameter values for the scoring function.

**Reasons To Reject:**

While the paper's explicit focus is on soft prompts and prompt tuning, the experiments do not demonstrate the behavior of the affinity scoring function when using manually-written/engineered prompts in the prompt pool. The paper also does not justify why soft prompts are preferable to manual prompts in this particular setting. (For example, is there a fundamental incompatibility, or are soft prompts simply the authors' preference here?) Therefore, it is unclear how beneficial the affinity scoring function would be when given manually-written task prompts in the prompt pool, or a possible mix of manual and soft prompts. This is an issue that should be discussed in some way, for the sake of completeness.

**Reproducibility:**

4: Could mostly reproduce the results, but there may be some variation because of sample variance or minor variations in their interpretation of the protocol or method.

**Reviewer Confidence:**

3: Pretty sure, but there's a chance I missed something. Although I have a good feel for this area in general, I did not carefully check the paper's details, e.g., the math, experimental design, or novelty.

**Typos Grammar Style And Presentation Improvements:**

Line 237: benchmark --> benchmarks

Line 279-280: measurement --> measurements

Line 338: for --> of

Line 449: There's a space between "types" and the superscript 2.

Lines 605-613: It seems like the Ethics Statement section was left in by accident? The text is from the EMNLP template.

Line 1004: Missing a period after "Table 4".

---

> ### Author Rebuttal · Authors · 2023-08-27
>
> Thanks for your valuable suggestions!
>
> Manually-written prompts need carefully handcrafted design. Finding proper prompts for each task is difficult and time-consuming. Even if significant effort is invested, manual prompts are likely to be suboptimal[1][2]. So we use soft prompt in the experiments. We will add the comparison of soft prompts and manually-written prompts in the revised version.
>
> We answer your questions as follows.
>
> A）	Yes, It means that prompts effectively enable LLMs to access their knowledge. I will make it more clear in the revised vesion.
>
> B）	Yes, the initial prompt for each task consists of random tokens. Finding proper prompts for each task is difficult and time-consuming. Even if significant effort is invested, manual prompts are likely to be suboptimal[1][2]. So we initialize prompt for each task with random tokens.
>
> C）	Yes, it is an LLM. Specifically, we use RoBERTa-large as the task-specific model as mentioned in Section 4.2 (Line 250).
>
> D）	Traditional methods use data size and task type to predict task transferability. They take tasks with large data size or the same task type as the best source tasks to transfer. However, experimental results show that positive transfer can occur when the source task has a small size or a different task type. So it is not precise to use data size or domain similarity to estimate task affinity.
>
> E）	We report the average score of multiple runs in Table 1.
>
> [1] P-Tuning v2: Prompt Tuning Can Be Comparable to Fine-tuning Universally Across Scales and Tasks. Liu et al.,2021
>
> [2] Making Pre-trained Language Models Better Few-shot Learners. Gao et al.,2020.

---

### Official Review · Reviewer_uUvq · 2023-08-03

**Soundness:** 3

**Excitement:**

4: Strong: This paper deepens the understanding of some phenomenon or lowers the barriers to an existing research direction.

**Missing References:**

Vu et al. 2021 is cited out of place for prompt tuning (L 056).

**Paper Topic And Main Contributions:**

This paper proposes a method for predicting task transferrability by learning a scoring function based on soft prompts. They show that the transferrability identified applies both to transfer learning with soft prompts as well as to full intermediate fine-tuning.

**Reasons To Accept:**

The method is sound and provides a moderate improvement over existing methods for predicting transferrability using soft prompts. The extension to full fine-tuning is also novel (to my knowledge). The experiments could be more extensive, but large answer the given research question.

**Reasons To Reject:**

My primary concern is regarding the set of target tasks, which is 1) small (only 6) and 2) not greatly separate from the training tasks (e.g. Yelp Polarity is a sentiment classification task and among the training tasks, while the SST-2 is also a sentiment task and in the target tasks). The authors do address generalization more concretely in 4.6.2 on a small experiment. While the results point in the right direction, the task-type overlap for the remaining experiments do mean that their results need to be taken with a grain of salt. I would be willing to raise the score with more such OOD experiments.

**Reproducibility:**

4: Could mostly reproduce the results, but there may be some variation because of sample variance or minor variations in their interpretation of the protocol or method.

**Reviewer Confidence:**

5: Positive that my evaluation is correct. I read the paper very carefully and I am very familiar with related work.

**Typos Grammar Style And Presentation Improvements:**

I would emphasize up front that this work studies the same problem as Vu et al., 2021., and that the key novelty is using a learned scoring function.

---

> ### Author Rebuttal · Authors · 2023-08-27
>
> Thanks for your suggestion! Besides the OOD experiments conducted in 4.6.2, we also remove the NLI tasks from the prompt pool and use the remaining tasks to train the affinity scoring function. Then we test its prediction performance on the SciTail task (NLI task). Experimental results show that the largest transfer gain on the SciTail task is 3.16% in accuracy, which suggests the generalization of the affinity scoring function on new task types. We will add more such OOD experiments in the revised version.

---

### Meta-Review · Area_Chair_9SwV · 2023-09-15

**Recommendation:** 5

**Metareview:**

The paper introduces a novel method for predicting task transferability in the context of natural language understanding tasks. The approach involves learning an affinity scoring function based on soft prompts, allowing efficient selection of source tasks for transfer learning. Extensive experiments demonstrate the method's effectiveness compared to baselines, although some additional comparisons and clarifications are needed. The paper also highlights the practicality of using soft prompts over manually-crafted prompts and emphasizes its applicability to various NLU tasks.

The authors commit to including a comparison of soft prompts and manually-written prompts, as well as more OOD experiments in the revised version. While they promise to look into LLMs like ChatGPT, I don't think that's necessary given that direct access to the weights is not given, making the approach difficult to assess from a research perspective.

---

### Decision · Program_Chairs · 2023-10-07

**Decision:**

Accept-Main

**Comment:**

The paper introduces a novel method for predicting task transferability in the context of natural language understanding tasks. The approach involves learning an affinity scoring function based on soft prompts, allowing efficient selection of source tasks for transfer learning. Extensive experiments demonstrate the method's effectiveness compared to baselines, although some additional comparisons and clarifications are needed. The paper also highlights the practicality of using soft prompts over manually-crafted prompts and emphasizes its applicability to various NLU tasks.

The authors commit to including a comparison of soft prompts and manually-written prompts, as well as more OOD experiments in the revised version. While they promise to look into LLMs like ChatGPT, I don't think that's necessary given that direct access to the weights is not given, making the approach difficult to assess from a research perspective.